# RARe: Retrieval-Augmented Retrieval With In-Context Examples

**Atula Tejaswi♠, Yoonsang Lee♡, Sujay Sanghavi♠\*, Eunsol Choi◇\***
♠The University of Texas at Austin  ♡Princeton University  ◇New York University
atutej@utexas.edu

## Abstract

While in-context learning is well-studied with decoder-only language models (LLMs), its utility for encoder-only models remains underexplored. We study in-context learning for encoder-only models for text retrieval tasks. Can incorporating in-context examples (query-document pairs) to the target query enhance retriever performance? Our approach, RARe, finetunes a pre-trained model with in-context examples whose query is semantically similar to the target query. This approach achieves performance gains of up to +2.72% nDCG across open-domain retrieval datasets (BeIR, RAR-b) compared to using the target query only as an input. In particular, we find RARe exhibits stronger out-of-domain generalization compared to models using queries without in-context examples, similar to what is seen for in-context learning in LLMs. We further provide analysis on the design choices of in-context example augmentation for retrievers and lay the foundation for future work.

## 1 Introduction

In-context learning (ICL) (Brown et al., 2020) has emerged as a powerful paradigm enabling diverse applications without parameter updates in large language models (LLMs). While in-context learning has been extensively studied for the decoder-only models (Xu et al., 2023; Min et al., 2022a; Dong et al., 2024), its application to encoder-only, embedding models have been limited. Recently, Li et al. (2024) proposed to incorporate randomly sampled few-shot examples into the query side to enhance the query embedding for embedding models. They reported strong performances, achieving state-of-the-art performances in a popular retrieval benchmark (Muennighoff et al., 2023). In this paper, we further study how in-context examples enhance the performance in retriever models, exploring its design choices and generalization capability with controlled studies.

Compared to decoder models which generates new tokens from the input as a continuation, how in-context examples would impact the output vector from an embedding model is less obvious. Unlike in decoder-only LLMs where in-context examples expand model capacity at generation time, in-context examples in encoder-only models may primarily provide task-relevant information rather than increasing model capacity. We study injecting *semantically similar* in-context examples to build a dense retriever model (Karpukhin et al., 2020) which embeds queries and documents into a shared representational space for efficient search over a large corpus. State-of-the-art retriever models started to leverage decoder-only models as a backbone (Wang et al., 2024b; BehnamGhader et al., 2024; Muennighoff et al., 2024; Meng et al., 2024; Lee et al., 2024a), further motivating our study of applying in-context examples.

We begin by naively prepending in-context examples to the target query and provide it to existing retriever models (BehnamGhader et al., 2024; Wang et al., 2024b; Meng et al., 2024). Unlike in decoder-only models, zero-shot modification leads to significant performance drop. We propose to construct retrieval models that can effectively leverage in-context examples, which we name as RARe: Retrieval Augmented Retrieval with In-Context

---

\*Equal advising.

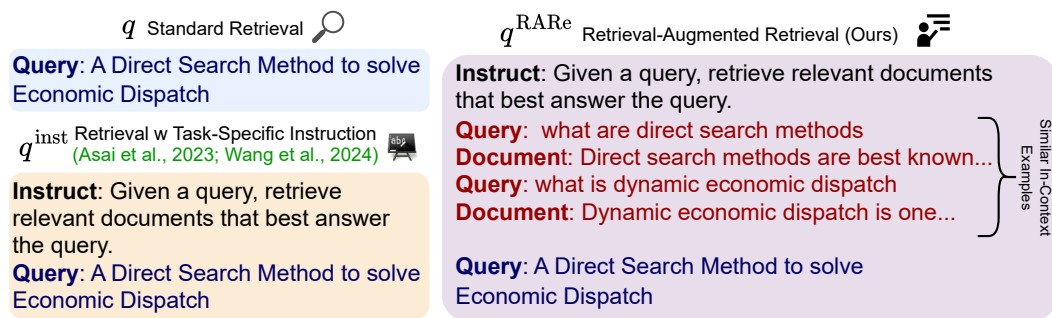

Figure 1: Overview. Prior work augments a task-specific instruction to a given query as input to the Retriever. In RARe, we further leverage a set of in-context exemplars that contain pairs of queries and relevant documents. These in-context examples are augmented with the original query as input to the retriever along with the instruction.

**Examples.** Our approach modifies the query format of retrieval systems by providing in-context examples whose query is semantically similar to the target query. Then, we apply standard continued fine-tuning with contrastive loss. We conduct a comprehensive evaluation of new query format across various experimental settings, initializing from both decoder-only checkpoints and pre-trained retriever model checkpoints. We demonstrate that RARe outperforms baseline models across multiple tasks, achieving improvements of up to +1.41% nDCG@10 compared to methods that do not use in-context examples on standard retrieval benchmarks (Thakur et al., 2021), and showing even larger gains (+2.72% nDCG@10) on reasoning-oriented retrieval tasks (Xiao et al., 2024).

Our contributions can be summarized as follows:

- We introduce RARe, an approach that adapts pre-trained checkpoints to use semantically similar in-context examples for retrieval.
- We demonstrate that this recipe can improve the performance of various base architectures, including decoder-only models and existing retriever models.
- We provide detailed analyses on how the quality, quantity, and selection of in-context examples affect performance, contextualizing the sources of performance gains in embedding-based ICL.

All our code and model checkpoints are publicly released.[1]

## 2 Method

**Standard Retrieval Setting** We consider a dense retriever (Karpukhin et al., 2020), where input queries $q$ and documents $d$ are encoded with an embedder $E(\cdot)$ into a fixed-dimensional embedding. The embedder $E(\cdot)$ is trained on a training set $\mathcal{D}$ which consists of multiple retrieval tasks $\{\mathcal{D}_1, \mathcal{D}_2, \cdots, \mathcal{D}_T\}$, where each task contains training examples of the form $(q, d^+, d^-)$ (Wang et al., 2024b; BehnamGhader et al., 2024). Here, $q$ is the input query, $d^+$ is a positive (relevant) document, and $d^-$ is a hard-negative (irrelevant) document, which allows for a contrastive-loss based training.

The evaluation task $\mathcal{D}_{\text{test}}$ consists of a corpus of documents $C$, as well as test pairs $(q, R^+)$, where $R^+ = \{d_1^+, d_2^+, ..., d_m^+\} \subset C$ is a set of relevant document(s) for the query (Thakur et al., 2021). The aim is to retrieve these relevant documents $R^+$ from the corpus $C$ using the embedder $E(\cdot)$. Specifically, an index $C_e$ of the corpus with document embeddings $E(d), \forall d \in C$ is created. Then, the embedding $E(q)$ of a test query $q$ is used to retrieve the documents $d$ whose embedding $E(d)$ is closest to $E(q)$, typically with the cosine (cos) similarity function.

---

[1] https://github.com/atutej/RARe

---

**Algorithm 1:** RARe - Training

---

**Input:** Training set $\mathcal{D}$, embedder $E(\cdot)$, BM25, the number of in-context examples $k$, mini-batch size $B$.

1: **for** each training iteration **do**
2:     Sample mini-batch $\mathcal{B}$ of size $B$ from $\mathcal{D}$
3:     **for** $(t_i, q, d^+, d^-) \in \mathcal{B}$ **do**
4:         In-Context Example Retrieval:
5:         $\{q_1^{\text{RARe}}, q_2^{\text{RARe}}, \ldots, q_k^{\text{RARe}}\} \leftarrow$ Retrieve nearest queries of $q$ from $\mathcal{D}$ using BM25
6:         $\{d_1^{\text{RARe+}}, d_2^{\text{RARe+}}, \ldots, d_k^{\text{RARe+}}\} \leftarrow \{d^+ : (q', d^+) \in \mathcal{D}, q' \in \{q_1^{\text{RARe}}, \ldots, q_k^{\text{RARe}}\}\}$
7:         $\mathcal{D}_i^{\text{RARe}} \leftarrow \{(q_1^{\text{RARe}}, d_1^{\text{RARe+}}), \ldots, (q_k^{\text{RARe}}, d_k^{\text{RARe+}})\}$
8:         Query Augmentation:
9:         $q^{\text{RARe}} = $ Instruct: $\{t_i\}$; Query: $\{q_1^{\text{RARe}}\}$; Document: $\{d_1^{\text{RARe+}}\} \cdots$; Query: $\{q\}$
10:    Training with Contrastive Loss:
11:    Compute the mini-batch contrastive loss $\mathcal{L}_{\text{in-context}}$ as described in Equation 5.
12:    Update $E(\cdot)$ by minimizing $\mathcal{L}_{\text{in-context}}$.

**Output:** Trained embedder $E(\cdot)$

---

**Retrieval with Instruction Setting**   Current architectures (Asai et al., 2023; BehnamGhader et al., 2024) prepend task-specific instruction $t_i$, $i \in [1, 2, \cdots, T]$ to the query to contextualize the task:

$$q^{\text{inst}} = \text{Instruct: } \{t_i\}; \text{ Query: } \{q\}, \quad q \in \mathcal{D}_i \tag{1}$$

Then, the embedder $E(\cdot)$ is trained with a standard contrastive loss (Izacard et al., 2022; Karpukhin et al., 2020), incorporating $q^{\text{inst}}$, and $d^+, d^- \in \mathcal{D}_i$, along in-batch negatives $n \in \mathbf{N}$, where $\mathbf{N}$ represents the set of in-batch negatives,

$$e_{q^{\text{inst}}} = E(q^{\text{inst}}); \quad e_{d^+} = E(d^+); \quad e_{d^-} = E(d^-); \quad e_n = E(n) \tag{2}$$

$$\mathcal{L} = -\log \frac{f(e_{q^{\text{inst}}}, e_{d^+})}{f(e_{q^{\text{inst}}}, e_{d^+}) + f(e_{q^{\text{inst}}}, e_{d^-}) + \sum_{n \in \mathbf{N}} f(e_{q^{\text{inst}}}, e_n)} \tag{3}$$

Where $f(x, y) = \exp(\cos(x, y))$. During evaluation on $\mathcal{D}_{\text{test}}$, each test query is augmented with task-specific instruction $t_{\text{test}}$.

**BGE-ICL Setting**   Recently, Li et al. (2024) proposed BGE-ICL, a model that augments in-context examples to enhance the query representation of retrievers. Specifically, given a query $q$, an in-context example set $\mathcal{D}_i^{\text{ic}} = \{(q_1^{\text{ic}}, d_1^{\text{ic+}}), (q_2^{\text{ic}}, d_2^{\text{ic+}}), \cdots, (q_k^{\text{ic}}, d_k^{\text{ic+}})\}$ is constructed by *randomly sampling* from $\mathcal{D}_i \in \mathcal{D}$, which are then augmented into the original query $q^{\text{inst}}$ to obtain $q^{\text{inst+ic}}$

$$q^{\text{inst+ic}} = \text{"Instruct: } \{t_i\}; \text{ Query: } \{q_1^{\text{ic}}\}; \text{ Document: } \{d_1^{\text{ic+}}\} \cdots; \text{ Query: } \{q\}\text{"} \tag{4}$$

Then, $E(\cdot)$ is trained with the same loss as Equation 3, but with $q^{\text{inst+ic}}$ instead of $q^{\text{inst}}$,

$$\mathcal{L}_{\text{in-context}} = -\log \frac{f(e_{q^{\text{inst+ic}}}, e_{d^+})}{f(e_{q^{\text{inst+ic}}}, e_{d^+}) + f(e_{q^{\text{inst+ic}}}, e_{d^-}) + \sum_{n \in \mathbf{N}} f(e_{q^{\text{inst+ic}}}, e_n)} \tag{5}$$

**Our RARe Setting**   We propose to enhance the query representation by incorporating *semantically-similar* in-context examples. This provides additional, highly relevant query-specific guidance to the model.

Given a query $q$, we use BM25 (Robertson & Zaragoza, 2009), a sparse retriever that ranks documents based on keyword matching, and find $k$ closest queries $q_j$ from $\mathcal{D}_i \in \mathcal{D}$ to obtain in-context examples $\mathcal{D}_i^{\text{RARe}} = \{(q_1^{\text{RARe}}, d_1^{\text{RARe+}}), (q_2^{\text{RARe}}, d_2^{\text{RARe+}}), \cdots, (q_k^{\text{RARe}}, d_k^{\text{RARe+}})\}$. As shown in Figure 1, we augment these examples to the original query $q^{\text{inst}}$ to obtain $q^{\text{RARe}}$,

$$q^{\text{RARe}} = \text{``Instruct: } \{t_i\}; \text{ Query: } \{q_1^{\text{RARe}}\}; \text{ Document: } \{d_1^{\text{RARe+}}\} \cdots; \text{ Query: } \{q\}\text{''} \tag{6}$$

We then train $E(\cdot)$ using Equation (5) to leverage these semantically similar examples. Algorithm 1 presents our training procedure in detail. At inference time, we similarly perform a search to find nearest in-context examples to form an augmented query. Algorithm 2 in the Appendix provides an overview of the inference procedure.

## 3 Experimental Setup

### 3.1 Fine-Tuning

**Base Models** We explore two training setups: fine-tuning decoder-only models for retrieval, and fine-tuning existing retriever models. For the first setup, we train the *Llama-3* family of models, following the training methodology outlined by Ma et al. (2023); Weller et al. (2024b). For the second setup, we use two high-performing publicly available embedding models that were trained with task-specific instructions: *LLM2Vec-Llama-3-8b-Supervised* (BehnamGhader et al., 2024) and *E5-Mistral-Instruct* (Wang et al., 2024b). We chose these two models because, unlike some other strong performers (Meng et al., 2024; de Souza P. Moreira et al., 2024), they were not trained on most of the datasets used in our downstream benchmarks. The *LLM2Vec-Llama-3-8b-Supervised* model is initially trained using an unsupervised text reconstruction objective and then fine-tuned with supervised contrastive learning on a public subset of the E5 dataset, which incorporates various supervised training datasets (Gao et al., 2021; Nguyen et al., 2016; Kwiatkowski et al., 2019). In contrast, *E5-Mistral-Instruct* undergoes further training on synthetic data that is not publicly available. These models are chosen to assess the impact of additional supervised training on an existing retriever model versus training a model for retrieval from scratch.

**Training Data** For fine-tuning existing retriever models, we follow prior work (BehnamGhader et al., 2024) and train on a publicly available portion of E5 dataset (Springer et al., 2024; Wang et al., 2024b), which contains MS-MARCO (Nguyen et al., 2016) NLI (Gao et al., 2021), ELI5 (Fan et al., 2019), FEVER (Thorne et al., 2018), HotpotQA (Yang et al., 2018), NQ (Kwiatkowski et al., 2019), SQuAD (Rajpurkar et al., 2016), Quora Duplication Questions (DataCanary et al., 2017), TriviaQA (Joshi et al., 2017). For fine-tuning models from LLM checkpoint, we use the MS-MARCO (Nguyen et al., 2016) passage ranking dataset and train without an instruction prefix, following Ma et al. (2023).

**Constructing In-Context Examples** During training, we provide each training example with five in-context examples from the dataset that it belongs to ($k$=5). Specifically, the set of examples $\mathcal{D}_i^{\text{RARe}}$ for each task is drawn from the training set $\mathcal{D}_i$, $q \notin \mathcal{D}_i^{\text{RARe}}$.

### 3.2 Evaluation

**Datasets** We evaluate on the widely used BeIR retrieval benchmark (Thakur et al., 2021). For ablative experiments, we follow prior work and focus on low-resource datasets (Wang et al., 2023) that potentially benefit more from few-shot examples. Since the BeIR benchmark contains a few datasets whose training sets are in the E5 dataset mixutre, we categorize them as in-domain and out-of-domain (i.e. datasets not seen during training). See Table 6 in the Appendix for a list of in-domain and out-of-domain datasets from BeIR. We also evaluate on a subset of the RAR-b (Xiao et al., 2024) benchmark, which requires complex reasoning for retrievers. Specifically, we evaluate on HellaSwag (Zellers et al., 2019), PIQA (Bisk et al., 2020), ARC-C (Clark et al., 2018), TempReason-L1 (Tan et al., 2023), WinoGrande (Sakaguchi et al., 2021), $\alpha$-NLI (Bhagavatula et al., 2020), SiQA (Sap et al., 2019), and Quail (Rogers

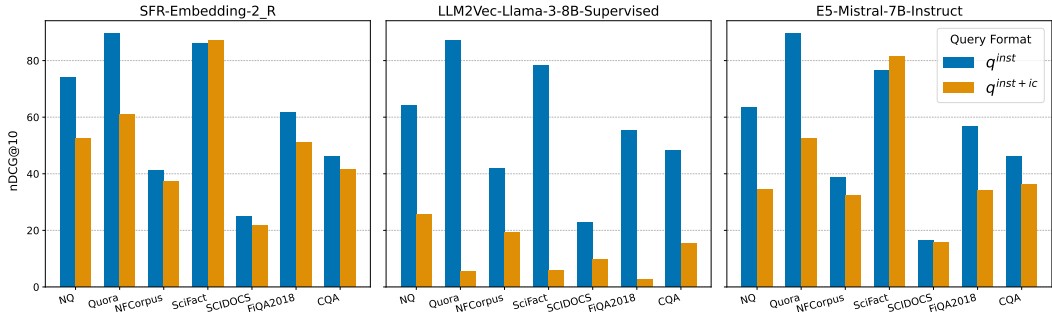

Figure 2: **Inference-only modification does not work.** We report performance before and after adding in-context examples to the query without updating model parameters. Embedding models are not able to leverage in-context examples out of the box, as opposed to decoder-only models.

et al., 2020). Unlike BeIR, some RAR-b queries are composed of sentences with (multiple) indicators (e.g., Start:, End:). Each dataset is associated with a task-specific instruction, following prior work (Muennighoff et al., 2023; Wang et al., 2024b; BehnamGhader et al., 2024). We provide additional preprocessing details in Appendix A.

**Constructing In-Context Examples** We construct $\mathcal{D}_{\text{test}}^{\text{RARe}}$ from the training/development set of each datasets. For datasets on BeIR that do not have either of these, we use a synthetically generated collection of document-query pairs (GenQ) from Thakur et al. (2021). For all experiments, we use $k$=5 in-context examples.

**Metrics** We use standard metrics for retrieval benchmarks. Following Thakur et al. (2021), we report nDCG@10, which measures the ranking quality of the top 10 retrieved documents, taking into account both the relevance and position of each retrieved document.

# 4 Results

We evaluate in-context example augmented queries in three settings. First, we evaluate the performance after inference-only modification, where we take existing pre-trained retrievers and simply provide in-context examples at inference time (Section 4). Second, we evaluate training retriever with in-context examples from an LLM (decoder-only) backbone (Section 4.1). Third, we compare training retriever models with in-context examples from a pre-trained retriever (Section 4.2).

**Inference-only Modification** Figure 2 illustrates the impact of incorporating in-context examples at inference time. Here, we simply modify the query format with retrieved in-context examples (i.e. $q^{\text{RARe}}$, Eq. 6) at inference time and compare its performance with the query format that does not have retrieved in-context examples (i.e. $q^{\text{inst}}$, Eq. 1). We evaluate the performance on three retriever models: *SFR-Embedding-2-R* (Meng et al., 2024), *LLM2Vec-Llama-3-8B-Supervised* (BehnamGhader et al., 2024), and *E5-Mistral-7B-Instruct* (Wang et al., 2024b). Unlike in autoregressive LLMs, these embedding models generally exhibit decreased performance when in-context examples are added, with *LLM2Vec-Llama-3-8B-Supervised* showing the largest drops in performance, except on one dataset (SciFact), where 2 out of 3 models show marginal gains over providing only instructions. Our experiments, which include adding more in-context examples and using nearest-neighbor examples, extend the findings of Muennighoff et al. (2024), where in-context examples led to decrease in performance on the GritLM models.

Table 1: **Training from decoder-only (LLM) checkpoint.** We initialize all models with *Llama-3.1-8b-Instruct*. Performance is measured by nDCG@10. RARe shows +2.72% and +0.87% nDCG@10 over Promptriever and ICL, respectively, on the reasoning oriented RAR-b benchmark. We provide a breakdown of In-Domain (ID), Out-of-Domain (OOD), and overall (Average) performance.

| Method | Training Data | ID | OOD | | Average |
|---|---|---|---|---|---|
| | | BeIR | BeIR | RAR-b | |
| RepLLaMA | MS-MARCO | **43.67** | 54.34 | 19.20 | 39.07 |
| Promptriever | MS-MARCO + Synthetic | 42.70 | **56.10** | 20.95 | 39.94 |
| ICL | MS-MARCO | 42.57 | 53.91 | 22.80 | 39.76 |
| RARe | MS-MARCO | 42.93 | 56.05 | **23.67** | **40.88** |

Table 2: **Training from retriever checkpoints.** Performance (nDCG@10) on BeIR (Thakur et al., 2021) and RAR-b (Xiao et al., 2024) benchmarks when fine-tuning retriever model on E5 dataset. We report a breakdown of performance on In-Domain (ID), Out-of-Domain (OOD, and overall (Average) performance.

| Method | *LLM2Vec-Llama-3-8b-Supervised* | | | | *E5-Mistral-Instruct* | | | |
|---|---|---|---|---|---|---|---|---|
| | ID | OOD | | Average | ID | OOD | | Average |
| | BeIR | BeIR | RAR-b | | BeIR | BeIR | RAR-b | |
| Base | 71.31 | 49.28 | 21.55 | 47.38 | 71.95 | 49.33 | 22.17 | 47.81 |
| Instruct | 70.46 | 47.79 | **23.44** | 47.23 | 72.91 | 48.98 | 24.12 | 48.67 |
| ICL | **71.81** | 47.69 | 21.94 | 47.14 | 72.03 | 49.46 | 24.69 | 48.72 |
| RARe | 71.67 | **49.30** | 23.10 | **48.02** | **72.98** | **50.93** | **25.79** | **49.90** |

## 4.1 Training from LLM Checkpoints

Next, we present the results of applying our approach when training from LLM checkpoint. This might preserve in-context learning capacity of the LLM, which can be lost during standard IR training, which compresses query and passage into a fixed dimensional vector. We initialize our models with *Llama-3.1-8b-Instruct* (Dubey et al., 2024) to enable comparison with prior work Ma et al. (2023); Weller et al. (2024b).

**Comparison Systems**   We compare training with our in-context example augmented query with two baselines. The first baseline is vanilla query (Eq. 1), which was explored in RepLLaMA (Ma et al., 2023). The second baseline is Promptriever (Weller et al., 2024b) which augments query-specific instructions using a synthetically generated training set from MS-MARCO. The third baseline is ICL, which prepends random in-context examples to the query during training and inference, equivalent to that of Li et al. (2024). In all these systems, the task-specific instruction is a null string (Ma et al., 2023) as we train on a single task (MS-MARCO).

**Results**   Table 1 presents the performance on downstream benchmarks when training from LLM checkpoints. Comparing within the same base LLM checkpoint, our apporach outperforms both baselines (RepLLaMA and Promptriever). Our performance is competitive to that of Promptriever (Weller et al., 2024b), without incorporating synthetic data during training. Specifically, RARe achieves +0.94% nDCG@10 over Promptriever, and +1.12% nDCG@10 over random ICL on average.

## 4.2 Training from Retriever Checkpoints

Lastly, we continue training retriever models – *LLM2Vec-Llama-3-8B-Supervised* (BehnamGhader et al., 2024), *E5-Mistral-Instruct* (Wang et al., 2024b) on a training

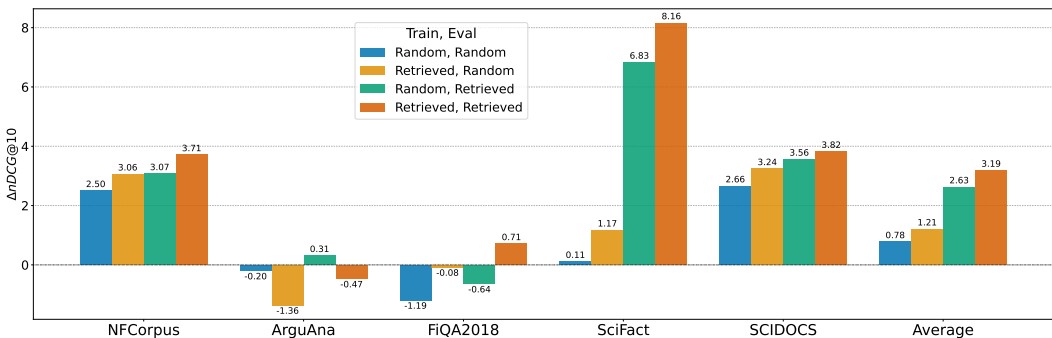

Figure 3: **Retrieved vs. Random In-context Examples.** Change in performance ($\Delta$nDCG@10) on *E5-Mistral-Instruct* with RARe ($q^{\text{RARe}}$) from the baseline setting ($q^{\text{inst}}$ both during training and evaluation time). Using retrieved examples during training and inference enhance model performance in most benchmark datasets.

set where queries are augmented with in-context examples. As these initial checkpoints have already been trained on the training dataset, the extent that retrievers adapt to new query format can be limited.

**Comparison Systems** We first report the initial retriever performance (**Base**) without any modification. Then, we compare continued fine-tuning with the task-specific instruction query format (Eq. 1) which only prepends the task specific instruction (**Instruct**, $q^{\text{inst}}$). Finally, we compare against **ICL** (Li et al., 2024), which augments random in-context examples to the query ($q^{\text{inst+ic}}$, Eq. 4) during training and evaluation.

**Results** Table 2 reports experimental results in this setting. Overall, both fine-tuning approaches provides gains over the base checkpoints. Comparing two settings, Instruct ($q^{\text{inst}}$) vs. RARe ($q^{\text{RARe}}$), our method achieves notable improvement with *E5-Mistral-Instruct* base model (up to +1.47% over random ICL on out-of-domain tasks, and +1.18% overall). Both RARe and ICL perform similar to Instruct ($q^{\text{inst}}$) setting when trained with the LLM2Vec base model. It is hard to attribute why experimental results varies based on the base retriever checkpoint, but we note the following differences between the two models. *LLM2Vec-Llama-3-8b-Supervised* is the only model in our experiments where further fine-tuning with only instructions led to a decrease in in-domain performance. *E5-Mistral-Instruct* employs causal attention with last token pooling, and trains on a proprietary synthetic dataset, *LLM2Vec-Llama-3-8b-Supervised* uses bidirectional attention with mean pooling, training only on the public portion.

The effectiveness of learning with in-context examples may depend on factors such as the underlying model architecture and data setting. Future work can further investigate the impact of base model characteristics, including architectural choices and other constraints.

## 5 Discussions and Analysis

### 5.1 Choice of In-context Examples

**Retrieved (Similar) vs. Random In-Context Examples** In Figure 3, we study the impact of retrieving the nearest neighbor query-document pairs as examples against randomly chosen examples during training and evaluation. We observe that using retrieved examples during both training and evaluation (Retrieved, Retrieved) consistently outperforms other configurations across most datasets. (Random, Retrieved) and (Retrieved, Random) generally outperform (Random, Random), suggesting retrieved examples are advantageous even when trained with randomly paired in-context examples. On ArguAna, we observe that (Retrieved, Random) performs the worst. There is a mismatch in the lengths of the queries

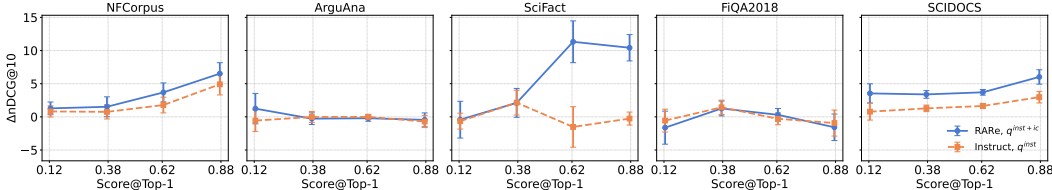

Figure 4: Change in performance ($\Delta$nDCG@10) from the base model (*E5-Mistral-Instruct*) for varying similarity between the closest in-context example query and target query (Score@Top-1). For RARe, we use retrieved in-context examples $q^{\text{RARe}}$ on the augmented in-context query format $q^{\text{inst+ic}}$.

Table 3: **Impact of the number of in-context examples (*k*) during training and evaluation.** All results are on *E5-Mistral-Instruct*. In general, performance increases when increasing the number of examples, and the optimal number of examples depends on the task.

| k | Arguana | CQADupStack | FiQA2018 | NFCorpus | SciFact | Touche2020 | Average |
|---|---|---|---|---|---|---|---|
| Instruct (0) | 61.19 | 44.82 | **57.39** | 40.99 | 77.28 | 29.35 | 51.84 |
| 1 | 60.47 | 46.76 | 56.07 | 40.67 | 81.47 | **29.78** | 52.54 |
| 3 | **62.98** | 47.12 | 57.08 | 40.77 | 83.71 | 27.12 | 53.13 |
| 5 | 60.87 | 48.46 | 57.31 | **42.28** | 84.79 | 28.70 | 53.74 |
| 10 | 58.85 | **48.92** | 57.03 | 42.24 | **87.61** | 28.29 | **53.82** |

used as in-context examples[2] (which are significantly shorter) versus the actual test queries in this dataset. This mismatch may introduce variability in performance, which has also been observed in decoder-only LLMs (Mishra et al., 2022). Overall, our findings align with prior work in in-context learning – that the incorporation of semantically similar examples is beneficial (Agrawal et al., 2022; Rubin et al., 2022). We observe similar overall trends on the other OOD datasets on the BeIR benchmark, reported in Figure 5 in the Appendix.

**Does Having Semantically Relevant In-Context Example Help?**  For some test examples, augmented in-context examples are very relevant, and for others, much less so. In this section, we group the evaluation examples by the maximum similarity of in-context query and the test query measured by an off-the-shelf sentence embedding model (Score@Top-1).[3] and plot the performances for each group. Figure 4 presents the performance of our system (RARe) and baseline (Instruct). On NFCorpus and SciFact datasets, we observe that when the closest in-context example has a high similarity with the target query, RARe demonstrates over 10% gains compared to the base model. On the other hand, fine-tuning with with $q^{\text{inst}}$ exhibits relatively lower performance gains with increasing similarity thresholds. In some datasets, such as ArguAna and FiQA2018, gains with increasing Score@Top-1 are less pronounced, but generally matches the performance of the base model. We observe similar overall trends on other BeIR OOD datasets (Figure 6 in the Appendix).

**How Many In-Context Examples Are Sufficient?**  We analyze the performance of RARe when varying the number of in-context examples provided during training and inference. Table 3 shows that increasing the number of in-context examples generally enhances performance. On ArguAna, we observe that 0 examples are optimal, which is likely due to the mismatch in the lengths of the queries used as in-context examples versus the actual test queries. The impact of adding more in-context example is not uniformly positive across all datasets, suggesting that the optimal number of in-context examples may be dataset-dependent. We observe similar trends when we fix the number of in-context examples to

---

[2]https://huggingface.co/datasets/BeIR/arguana-generated-queries

[3]https://huggingface.co/sentence-transformers/all-MiniLM-L6-v2

Table 4: **In-Context Format** Comparing variants of in-context example format on *E5-Mistral-Instruct*. Instruct refers to the baseline which does not use any in-context examples.

| Method | ArguAna | CQA | FiQA2018 | NFCorpus | SciFact | Touche2020 | Average |
|---|---|---|---|---|---|---|---|
| Instruct | **61.19** | 44.82 | **57.39** | 40.99 | 77.28 | **29.35** | 51.83 |
| Queries-Only | 58.88 | 46.66 | 54.44 | 41.42 | 78.84 | 28.09 | 51.39 |
| Doc-Only | 57.54 | 48.28 | 56.02 | 41.62 | 79.80 | 29.01 | 52.05 |
| Shuffle-NC | 60.17 | 45.78 | 54.25 | 41.17 | 80.70 | 29.18 | 51.88 |
| Shuffle-C | 58.97 | 47.97 | 55.98 | 41.78 | 80.51 | 28.97 | 52.36 |
| Regular | 60.87 | **48.46** | 57.31 | **42.28** | **84.79** | 28.70 | **53.74** |

Table 5: **Impact of adding negative documents in the in-context prompt.** All results are on *E5-Mistral-Instruct*. Negative documents ($d^-$) in the prompt do not enhance performance.

| Training / Eval Setting | ArguAna | CQA | FiQA2018 | NFCorpus | SciFact | Touche2020 | Average |
|---|---|---|---|---|---|---|---|
| RARe-$q^{\text{RARe}}$ | 60.87 | 48.46 | 57.31 | 42.28 | 84.79 | 28.70 | 53.74 |
| RARe-$q^{\text{RARe+neg}}$ | 61.19 | 48.09 | 56.89 | 41.58 | 82.37 | 30.51 | 53.44 |

five during training and vary the number of examples provided during inference, which are provided in Table 11 in the Appendix.

**Ablating Content and Format of In-context Examples** One can view in-context examples as a form of query expansion (Lv & Zhai, 2009; Wang et al., 2023), providing useful keywords to improve the performance. In Table 4, we analyze the impact of various formats of in-context examples. Each row represents a different model, fine-tuned with the format that they are evaluated on. Query-Only and Doc-Only contain only queries and documents of in-context examples, respectively. For Shuffle-Constrain (C), we randomly shuffle the mapping between $q$ and $d$. On the other hand, for Shuffle-No Constrain (NC), we do not assume any structure, meaning that a query can be followed by a query as well as a document.

First, we observe that Query-Only shows a larger performance drop over Doc-Only, suggesting in-context documents might contain more useful contents than in-context queries. Second, we observe that shuffling the pairings (Shuffle-C) marginally hurts in-context learning in RARe, as opposed to Shuffle-NC. Our findings align with prior study in decoder-only models (Min et al., 2022b) which showed strict correspondence between $q$ and $d$ is not required for performance gains from in-context examples. We observe similar trends when keeping the training format fixed (Regular) and vary only the evaluation format (Table 14 and Table 13 in the Appendix).

**Negative Documents in the Query** So far, we have used $(q, d^+)$ i.e (Query, Positive Document) pairs as the in-context prompt. Therefore, we study the impact of including negative documents. Specifically, the augmented query $q^{\text{RARe+neg}}$ includes examples of the form $(q, d^+, d^-)$, where the documents are prefixed with the term "Positive Document: " and "Negative Document: " respectively. Table 5 presents the downstream performance comparison between RARe variants trained solely on positive examples and those trained with augmented negative documents. The results indicate no performance gains from including negative documents. In fact, training with negative examples led to a slight decrease in performance.

## 6 Related Work

**In-context learning** ICL (Brown et al., 2020) allows models to adapt to new tasks in a few-shot manner by conditioning on the input data and the context provided at inference

time. ICL has been effectively applied to a wide range of tasks such as classification (Milios et al., 2023), translation (Zhu et al., 2024), mathematical reasoning (Wei et al., 2022; Zhou et al., 2022), and code generation (Li et al., 2023a). Recent advancements have enhanced the ICL capabilities of language models through additional training procedures (Huang et al., 2022; Gu et al., 2023; Shi et al., 2024). Min et al. (2022a) and Chen et al. (2022) perform meta-learning with in-context examples on a wide collection of tasks, with the goal of adapting to a new task at inference time through few-shot in-context examples. Other works have explored improving performance through more principled approaches to select in-context examples during inference (Zhang et al., 2022; Sorensen et al., 2022; Wang et al., 2024c; Qin et al., 2024; Lee et al., 2024b). A simple and popular approach is to retrieve examples that are most similar to the input (Liu et al., 2022; Rubin et al., 2022; Li et al., 2023c). Providing in-context examples to re-ranking models has been studied in prior work (Drozdov et al., 2023), but the potential of augmenting retrievers themselves by leveraging in-context examples remains unexplored. Muennighoff et al. (2024) explored providing an in-context example out-of-the-box, but showed an overall decrease in performance compared to zero-shot inference. The work most closely related to ours is by Li et al. (2024), who use a fixed, randomly selected set of in-context examples rather than retrieving query-specific ones. In contrast, our approach retrieves relevant examples tailored to each query and includes a comprehensive analysis of key design choices such as the quality, quantity, and format of in-context examples, as well as the choice of base model.

**Retrieval** Large language models pre-trained with autoregressive setups (Jiang et al., 2023; Dubey et al., 2024) have shown remarkable performance when adapted to retrieval tasks (Wang et al., 2024b; BehnamGhader et al., 2024), outperforming encoder-style retrievers (Izacard et al., 2022; Wang et al., 2024a). Despite these advancements, a challenge that remains is the ability to tailor retrieval systems to specific tasks or queries. To address this, a recent line of work explores incorporating instructions into retrieval by training models to use task-specific instructions along with the query (Su et al., 2023; Asai et al., 2023). Oh et al. (2024) and Weller et al. (2024a) further propose using instructions that are specific to each query. Another well-established technique in retrieval is query expansion (Jagerman et al., 2023; Li et al., 2023b; Chen et al., 2024), where the query is augmented with additional terms to enrich the embedding as a form of relevance feedback (Lv & Zhai, 2009). Recent efforts have focused on applying LLMs to expand the original query before retrieval (Wang et al., 2023; Shen et al., 2024; Dai et al., 2022). However, our focus is on the dense retrieval paradigm where a separate strong generative LLM is not available or used.

## 7 Conclusion

In this paper, we investigated the impact of in-context example augmentation for retrieval models, building on recent efforts to extend in-context learning for retrievers. We introduced RARe, a simple strategy that equips retrievers with the ability to leverage in-context examples by training with semantically similar in-context examples. Through detailed experiments and analyses, we demonstrated that RARe consistently improves performance across various architectures and downstream retrieval tasks, demonstrating the effectiveness of in-context learning for retriever models.

*Acknowledgments*

The work is partially supported by NSF grant IIS-2312948. This research has been supported by computing support on the Vista GPU Cluster through the Center for Generative AI (CGAI) and the Texas Advanced Computing Center (TACC) at UT Austin. We also thank Fangyuan Xu, Michael Zhang, Anuj Diwan, and other members of the UT NLP community for insightful feedback.

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

# Appendix

The appendix is organized as follows: In Appendix A, we present details on additional data preprocessing and other training details. In Appendix B, we present additional results and experiments.

## A   Experimental Details

### A.1   Training Details

**Hyperparameters**   For fine-tuning *Llama-3-8B*, we follow the setting outlined in Ma et al. (2023). We train on 4 H100 GPUs with per-device batch size 8 and gradient accumulation steps 4. We apply LoRA (Hu et al., 2021) with $r$=32, temperature of 0.01, learning rate 1e-4 with 100 warmup steps. We use a sequence length of 512 for documents and 1024 for queries as in-context augmented queries are longer. For RARe we use a mixture of 70% examples with in-context examples and 30% without Table 15.

When fine-tuning existing retriever models (*E5-Mistral-Instruct*, *LLM2Vec-Llama-3-8B-Supervised*), we follow a setting similar to BehnamGhader et al. (2024). We train on 8 H100 GPUs with a largest possible per-device batch size of 32 along with 2 gradient accumulation steps. We consider a random subset of 100*K* examples from the public E5 dataset mixture (Springer et al., 2024; Wang et al., 2024b). We use a learning rate of 2e-4, maximum sequence length 1024, warmup ratio 0.1 for 1 epoch. For *E5-Mistral-Instruct*, we apply LoRA (Hu et al., 2021) $r$=16, and $r$=4 for *LLM2Vec-Llama-3-8B-Supervised* since a higher rank was leading to severe overfitting on the instruction baseline.

### A.2   Data Processing

**RAR-b**   Since RAR-b benchmark provides only test split, we parse the original training data for each dataset to use as in-context examples. We exclude datasets without any training splits and 2 datasets that were a mixture of multiple tasks or datasets, thereby being difficult to parse. This results in 8 datasets to evaluate on. We preprocess the training split to match the format of RAR-b test split, without excluding any instances. An exception is made for $\alpha$-NLI, where there were multiple identical instances. Therefore, we removed such duplicates, resulting in 72,046 in-context candidates. Furthermore, some RAR-b queries are composed of sentences with (multiple) indicators (e.g., Start:, End:). To address this, we make a minor modification in formatting, enclosing the queries in brackets. The final query representation is:

$$q^{\text{RARe}} = \text{"Instruct: } \{t\}; \text{ Query: } [\{q_1^{\text{RARe}}\}]; \text{ Document: } \{d_1^{\text{RARe+}}\} \cdots; \quad \text{Query: } [\{q\}]\text{"} \quad (7)$$

**Inference Algorithm**   Algorithm 2 provides a detailed outline of inference with RARe.

**Promptriever**   Promptriever (Weller et al., 2024b) employs 10 different prompts and reports the highest score for each dataset. We apply the prompt that works the best (outperforms 5/15 datasets), which is as follows: `A document that meets these criteria is considered relevant, while a document that does not meet these criteria is considered non-relevant.`

**Synthetic In-Context Examples for BeIR**   For QuoraRetrieval, MSMARCO, DBPedia, FiQA-2018, NFCorpus, and SciFact, we utilize the original training splits without incorporating synthetically generated examples. For the remaining datasets—ArguAna, ClimateFEVER, FEVER, HotpotQA, Touche2020, TREC-COVID, CQADupStack, SCIDOCS, and NQ—we employ synthetic examples provided by the BEIR benchmark (e.g., ArguAna: https://huggingface.co/datasets/BeIR/arguana-generated-queries). For further details, please refer to our codebase at https://github.com/atutej/RARe.

---

**Algorithm 2:** RARe - Inference

---

**Input:** A list of test queries $D^{test}$, Corpus $C$, embedder $E(\cdot)$, the number of in-context
        examples $k$, Training dataset $\mathcal{D}^{\mathcal{T}}$, task instruction $t$.

1:   $C_e \leftarrow$ Construct document index as $E(d), \forall d \in C$.
2:   **for** $i \in [0, len(D^{test})]$ **do**
3:      $q = D^{test}[i]$
4:      In-Context Example Retrieval:
5:      $\{q_1^{RARe}, q_2^{RARe}, \ldots, q_k^{RARe}\} \leftarrow$ Retrieve nearest neighbor queries of $q$ from $\mathcal{D}^{\mathcal{T}}$ using
        BM25
6:      $\{d_1^{RARe+}, d_2^{RARe+}, \ldots, d_k^{RARe+}\} \leftarrow \{d^+ : (q', d^+) \in \mathcal{D}, q' \in \{q_1^{RARe}, \ldots, q_k^{RARe}\}\}$
7:      $\mathcal{D}^{RARe} \leftarrow \{(q_1^{RARe}, d_1^{RARe+}), \ldots, (q_k^{RARe}, d_k^{RARe+})\}$
8:      Query Augmentation / Encoding:
9:      $q^{RARe} =$ Instruct: $\{t\}$; Query: $\{q_1^{RARe}\}$; Document: $\{d_1^{RARe+}\} \cdots$; Query: $\{q\}$
10:     $e_q \leftarrow E(q_{test}^{RARe})$
11:     Prediction:
12:     $d = \text{argmax}_{d \in C} \exp(\cos(e_q, e_d))$
13:     $D_{pred}.\text{append}(d)$

**Output:** Predictions $D_{pred}$.

---

# B   Additional Experiments

Table 6: Performance (nDCG@10) on BeIR (Thakur et al., 2021) when fine-tuning retriever model on E5 dataset. We report a breakdown of performance on In-Domain (ID) and Out-of-Domain (OOD) tasks on BeIR. * indicates statistical significance over Instruct ($p < 0.05$) using the paired bootstrap test. For the Average score, we compute the overall p-value using Fisher's method.

| Category | Dataset | *LLM2Vec-Llama-3-8b-Supervised* | | | | *E5-Mistral-Instruct* | | | |
| | | Base | Instruct | RARe | | Base | Instruct | RARe | |
| | Eval Format | $q^{inst}$ | $q^{inst}$ | $q^{inst}$ | $q^{RARe}$ | $q^{inst}$ | $q^{inst}$ | $q^{inst}$ | $q^{RARe}$ |
|---|---|---|---|---|---|---|---|---|---|
| | FEVER | **90.20** | 88.12 | 88.43 | 86.62 | 87.84 | **91.50** | 90.18 | 90.48 |
| | HotpotQA | 71.76 | 72.50 | 73.83 | **79.09\*** | 75.72 | 73.91 | 72.18 | **75.95\*** |
| ID | NQ | 64.21 | 63.63 | 65.00 | **66.13\*** | 63.53 | 67.44 | **68.15** | 67.66 |
| | QuoraRetrieval | 87.16 | 87.85 | **87.88** | 87.63 | 89.61 | **89.82** | 89.59 | 88.95 |
| | MSMARCO | **43.24** | 40.19 | 40.77 | 38.88 | 43.06 | 41.89 | 41.88 | 41.88 |
| | ArguAna | **62.78** | 60.51 | 59.54 | 57.05 | 61.65 | 61.19 | **62.90** | 60.87 |
| | ClimateFEVER | 34.27 | 34.49 | 34.67 | **34.73\*** | 38.35 | **39.03** | 38.99 | 37.50 |
| | CQADupStack | 48.25 | 49.76 | 49.10 | **49.93** | 42.97 | 44.82 | 45.57 | **48.46\*** |
| | DBPedia | 48.34 | 48.61 | 48.41 | **49.09\*** | 48.89 | 48.92 | 49.24 | **49.65\*** |
| OOD | FiQA2018 | **55.33** | 52.99 | 54.26 | 52.82 | 56.81 | **57.39** | 56.33 | 57.31 |
| | NFCorpus | 41.83 | **41.92** | 41.61 | 41.84 | 38.58 | 40.99 | 41.19 | **42.28\*** |
| | SCIDOCS | 22.96 | **23.97** | 22.92 | 23.35 | 16.32 | 17.94 | 18.71 | **20.19\*** |
| | SciFact | 78.22 | 76.89 | 77.70 | **81.77\*** | 76.42 | 77.28 | 77.11 | **84.79\*** |
| | Touche2020 | 20.50 | 22.11 | **22.71** | 19.54 | 26.27 | **29.35** | 27.56 | 28.7 |
| | TRECCOVID | 80.34 | 68.37 | 78.55 | **82.78\*** | 87.03 | 72.89 | 77.03 | 79.58 |
| | **Average** | 56.63 | 55.35 | 56.36 | **56.76\*** | 56.87 | 56.96 | 57.11 | **58.28\*** |

Table 7: Performance on reasoning-focused IR benchmark RAR-b (Xiao et al., 2024) when fine-tuning existing retriever models. * indicates statistical significance over Instruct ($p < 0.05$) using the paired bootstrap test. For the Average score, we compute the overall p-value using Fisher's method.

| | LLM2Vec-Llama-3-8b-Supervised | | | | E5-Mistral-Instruct | | | |
| Dataset | Base | Instruct | RARe | | Base | Instruct | RARe | |
| Eval Format | $q^{inst}$ | $q^{inst}$ | $q^{inst}$ | $q^{RARe}$ | $q^{inst}$ | $q^{inst}$ | $q^{inst}$ | $q^{RARe}$ |
|---|---|---|---|---|---|---|---|---|
| ARC-C | **18.81** | 18.77 | 18.28 | 17.02 | 19.00 | 20.37 | 22.72 | **26.44*** |
| $\alpha$-NLI | 26.59 | **27.29** | 25.25 | 23.66 | **26.04** | 25.70 | 24.19 | 23.23 |
| HellaSwag | **34.32** | 34.19 | 34.19 | 33.29 | 35.38 | 35.99 | 35.07 | **36.29*** |
| PIQA | 33.57 | 37.07 | 38.12 | **39.72*** | 39.80 | 39.35 | 37.22 | **41.35*** |
| Quail | **6.83** | 6.06 | 5.57 | 4.25 | 8.40 | 10.94 | **15.34** | 14.69 |
| SiQA | **6.99** | 5.34 | 4.39 | 4.55 | **5.66** | 5.45 | 5.75 | **6.15** |
| TempReason-L1 | 5.24 | 5.89 | 5.55 | **7.87*** | 3.60 | **4.71** | 4.55 | 4.67 |
| WinoGrande | 40.02 | 52.88 | 48.47 | **54.44*** | 39.48 | 50.41 | 44.26 | **53.50*** |
| Average | 21.55 | **23.44** | 22.48 | 23.10 | 22.17 | 24.12 | 23.64 | **25.79*** |

Table 8: Performance (nDCG@10) on BeIR when training decoder-only models. * indicates statistical significance over RepLLaMA ($p < 0.05$) using the paired bootstrap test. For the Average score, we compute the overall p-value using Fisher's method.

| | Llama2 | Llama3 | | Llama-3.1-Instruct | | | |
| Dataset | RepLLaMA | RepLLaMA | RARe | RepLLaMA | Promptreiver | RARe | |
| Eval Format | $q^{inst}$ | $q^{inst}$ | $q^{RARe}$ | $q^{inst}$ | $q^{inst}$ | $q^{inst}$ | $q^{RARe}$ |
|---|---|---|---|---|---|---|---|
| ArguAna | 48.60 | 52.83 | 49.48 | 51.38 | 58.90 | **54.77** | 52.83 |
| ClimateFEVER | 29.30 | 32.52 | 32.12 | 33.13 | 29.80 | **35.91** | 34.24* |
| CQADupStack | 37.91 | 42.59 | 42.96 | 41.58 | 42.18 | 42.55 | **43.31*** |
| DBPedia | 44.80 | 45.62 | 45.79 | 44.73 | **46.00** | 45.87 | 45.95* |
| FEVER | 82.90 | 81.79 | 83.66 | 79.22 | **85.50** | 80.05 | 81.84* |
| FiQA2018 | 45.00 | 44.31 | 47.13 | 44.50 | **47.20** | 44.36 | 46.20* |
| HotpotQA | 68.80 | 72.24 | 72.72 | 70.90 | 71.70 | 70.55 | **74.01*** |
| MSMARCO | 42.00 | 43.56 | 44.77 | **43.67** | 42.70 | 41.65 | 42.93* |
| NFCorpus | 36.00 | 37.73 | 39.34 | 38.77 | 38.50 | 38.16 | **39.74*** |
| NQ | 63.00 | 62.70 | 65.96 | 61.09 | 63.80 | 60.92 | **65.20*** |
| Quora | 86.00 | 88.34 | 87.65 | 86.84 | 87.30 | **87.95** | 87.65* |
| SCIDOCS | 16.10 | 19.66 | 19.45 | 19.26 | **20.80** | 20.02 | 19.52 |
| SciFact | 75.30 | 75.02 | 77.20 | 75.38 | **77.50** | 74.59 | 76.54 |
| TRECCOVID | 83.90 | 83.15 | 85.76 | 83.15 | 84.50 | 77.52 | **85.30*** |
| Touche2020 | 34.10 | 27.84 | 32.89 | 30.77 | 31.70 | 25.47 | **32.38** |
| Average | 52.91 | 53.99 | 55.13 | 53.62 | **55.21** | 53.36 | 55.18* |

## B.1 Performance on BeIR and RAR-b

Table 6 and Table 7 provide detailed numbers on each dataset from BeIR and RAR-b respectively when training from retriever checkpoints. Table 8 and Table 9 provide detailed numbers on each dataset from BeIR and RAR-b respectively when training from decoder-only LLMs.

## B.2 Efficiency Evaluation

In Table 10, we present a breakdown of the latency of each stage of the retrieval pipeline for both baseline ($q^{inst}$) and in-context ($q^{inst+ic}$) settings. We measure the total time required to obtain nearest-neighbour in-context examples (**NN**) from BM25, compute query embeddings (**Query**), and perform search with FAISS (Douze et al., 2024) with encoded query embeddings on the pre-computed document index (**Search**). We observe that the largest

Table 9: Performance (nDCG@10) on datasets from RAR-b when training decoder-only models. * indicates statistical significance over Promptriever ($p < 0.05$) using the paired bootstrap test. For the Average score, we compute the overall p-value using Fisher's method.

| Dataset | Llama2 | Llama3 | | Llama-3.1-Instruct | | | |
| | RepLLaMA | RepLLaMA | RARe | RepLLaMA | Promptreiver | RARe | |
| Eval Format | $q^{inst}$ | $q^{inst}$ | $q^{RARe}$ | $q^{inst}$ | $q^{inst}$ | $q^{inst}$ | $q^{RARe}$ |
|---|---|---|---|---|---|---|---|
| ARC-C | 11.79 | 11.65 | 13.48 | 11.68 | 14.63 | 13.24 | **15.02** |
| $\alpha$-NLI | 25.40 | 24.35 | 30.38 | 24.96 | 24.70 | 27.34 | **31.58*** |
| HellaSwag | 30.83 | 31.47 | 30.27 | 31.03 | **32.57** | 31.42 | 28.81 |
| PIQA | 31.56 | 32.84 | 34.12 | 33.42 | 34.80 | 34.23 | **35.59*** |
| Quail | 6.40 | 6.21 | 5.98 | 5.71 | **7.80** | 6.92 | 6.91 |
| SiQA | 2.82 | 2.61 | **3.87** | 2.75 | 3.53 | 2.18 | 3.14 |
| TempReason-L1 | 1.49 | 1.75 | 3.61 | 2.05 | 4.32 | 4.84 | **6.59*** |
| WinoGrande | 51.58 | 37.11 | 57.01 | 42.01 | 45.25 | 44.72 | **61.69*** |
| **Average** | 20.23 | 18.50 | 22.34 | 19.20 | 20.95 | 20.61 | **23.67*** |

Table 10: Latency breakdown (in milliseconds per query) of each stage in the retrieval pipeline for $q^{inst}$ and $q^{RARe}$ evaluation settings. **# Corpus** denote the number of documents and **Avg Q len.** denote the average number of query tokens split by whitespace. Table 10 in the Appendix provides numbers on additional datasets.

| Dataset | # Corpus | Eval Setting | Avg Q len. | NN | Query | Search | Total | Inc. |
|---|---|---|---|---|---|---|---|---|
| NFCorpus | 3633 | $q^{inst}$ | 3.3 | 0 | 10.22 | 1.67 | 11.89 | - |
| | | $q^{RARe}$ | 866.0 | 0.62 | 473.65 | 1.76 | 476.04 | 40.04× |
| FiQA2018 | 57638 | $q^{inst}$ | 10.9 | 0 | 10.68 | 12.22 | 22.90 | - |
| | | $q^{RARe}$ | 1016.6 | 0.69 | 429.97 | 13.63 | 444.29 | 19.40× |
| TRECCOVID | 171332 | $q^{inst}$ | 10.6 | 0 | 36.60 | 81.60 | 118.20 | - |
| | | $q^{RARe}$ | 722.54 | 6.20 | 435.60 | 86.00 | 527.80 | 4.47× |
| Touche2020 | 382545 | $q^{inst}$ | 6.6 | 0 | 28.98 | 189.59 | 218.57 | - |
| | | $q^{RARe}$ | 1287.8 | 4.08 | 822.86 | 214.29 | 1041.22 | 4.76× |
| Quora | 522931 | $q^{inst}$ | 9.5 | 0 | 11.39 | 98.64 | 110.04 | - |
| | | $q^{RARe}$ | 129.5 | 0.32 | 53.03 | 98.26 | 151.61 | 1.38× |
| DBPedia | 4635922 | $q^{inst}$ | 5.5 | 0 | 92.33 | 1470.95 | 1563.28 | - |
| | | $q^{RARe}$ | 158.2 | 0.48 | 115.53 | 1773.18 | 1889.18 | 1.21× |
| SciFact | 5183 | $q^{inst}$ | 12.5 | 0 | 15.07 | 2.03 | 17.10 | - |
| | | $q^{RARe}$ | 1250.7 | 0.83 | 707.83 | 2.03 | 710.70 | 41.56× |
| SCIDOCS | 25657 | $q^{inst}$ | 9.4 | 0 | 11.29 | 5.74 | 17.03 | - |
| | | $q^{RARe}$ | 901.1 | 0.67 | 354.82 | 5.79 | 361.28 | 21.21× |
| CQADupStack | 38100 | $q^{inst}$ | 8.6 | 0 | 9.13 | 7.75 | 16.88 | - |
| | | $q^{RARe}$ | 678.2 | 1.15 | 466.23 | 7.79 | 475.17 | 28.15× |
| ClimateFEVER | 5416593 | $q^{inst}$ | 20.2 | 0 | 16.98 | 1124.36 | 1141.34 | - |
| | | $q^{RARe}$ | 831.3 | 2.31 | 424.60 | 1123.02 | 1549.93 | 1.36× |

contributing factors to latency are the average length of input queries (**Avg Q len.**), and the size of the index (**# Corpus**). The increased time for query encoding is not exclusive to our approach, since query expansion methods Wang et al. (2023); Drozdov et al. (2023) also require encoding longer queries. **Moreover, the increased latency can be viewed**

**as a form of test-time scaling, where the user trades-off throughput for performance depending on the preference. When no in-context examples are provided, we match the performance of the Instruct baseline**. For large query length and small corpus sizes, the in-context setting demonstrates a significant increase in total latency (19.40-40.04× for FiQA2018 and NFCorpus, respectively). However, for smaller average query lengths, this latency diminishes, as seen for Quora (1.38×) and DBPedia (1.21×). Moreover, the added latency due to the in-context setting also diminishes when the corpus size grows (which is the case in several real world settings), as the time required for search outweighs the time to encode the query. For example, on Touche2020 with a larger corpus of 380K documents, the increase in latency is 4.76× compared to FiQA2018 (19.40×) for similar query lengths. Note that we did not perform any optimization that can further reduce latency. For instance, when the in-context examples are fixed, their KV representations can be pre-computed and cached, providing latency that is comparable to when only the original query is available. Furthermore, future research could explore several promising avenues, such as using efficient long-context retrievers (Saad-Falcon et al., 2024; Zhang et al., 2024) as a backbone, or developing extractive and/or abstractive compression techniques of in-context examples.

Table 11: **Impact of the number of in-context examples (*k*) at inference time.** *k* = 5 during training. All results are on *E5-Mistral-Instruct*. In general, performance increases when increasing the number of examples, and the optimal number of examples can vary by task.

| | | # Examples | | | | |
|---|---|---|---|---|---|---|
| **Dataset** | **Instruct (0)** | **0** | **1** | **3** | **5** | **10** |
| ArguAna | 61.19 | **62.90** | 61.24 | 60.99 | 61.18 | 60.37 |
| ClimateFEVER | **39.03** | 38.99 | 38.27 | 37.97 | 37.50 | 37.67 |
| CQADupStack | 44.82 | 45.57 | 47.49 | 48.33 | 48.46 | **48.48** |
| DBPedia | 48.92 | 49.24 | 49.79 | 48.34 | 49.65 | **49.82** |
| FiQA2018 | 57.39 | 56.33 | **57.61** | 57.42 | 57.31 | 57.38 |
| NFCorpus | 40.99 | 41.19 | 41.48 | 42.10 | 42.28 | **42.29** |
| SCIDOCS | 17.94 | 18.71 | 19.83 | 20.17 | 20.19 | **20.20** |
| SciFact | 77.28 | 77.11 | 83.56 | 84.45 | 84.79 | **85.12** |
| Touche2020 | 29.35 | 27.56 | 27.53 | 27.70 | 28.70 | **30.77** |
| TRECCOVID | 72.89 | 77.03 | 76.96 | 78.99 | **79.58** | 78.77 |
| **Average** | 48.98 | 49.46 | 50.38 | 50.65 | 50.96 | **51.09** |

Table 12: **Impact of the number of in-context examples (*k*) during training and inference.** All results are on *E5-Mistral-Instruct*. In general, performance increases when increasing the number of examples, and the optimal number of in-context examples can vary by task.

| | | # Examples | | | | |
|---|---|---|---|---|---|---|
| **Dataset** | **Instruct (0)** | **0** | **1** | **3** | **5** | **10** |
| Arguana | 61.19 | **62.90** | 60.47 | 62.98 | 60.87 | 58.85 |
| ClimateFEVER | **39.03** | 38.99 | 37.94 | 36.45 | 37.50 | 36.54 |
| CQADupStack | 44.82 | 45.57 | 46.76 | 47.12 | 48.46 | **48.92** |
| DBPedia | 48.92 | 49.24 | 47.70 | 49.05 | **49.65** | 47.95 |
| FiQA2018 | **57.39** | 56.33 | 56.07 | 57.08 | 57.31 | 57.03 |
| NFCorpus | 40.99 | 41.19 | 40.67 | 40.77 | **42.28** | 42.24 |
| SCIDOCS | 17.94 | 18.71 | 20.01 | 19.28 | 20.19 | **21.54** |
| SciFact | 77.28 | 77.11 | 81.47 | 83.71 | 84.79 | **87.61** |
| Touche2020 | **29.35** | 27.56 | 29.78 | 27.12 | 28.70 | 28.29 |
| TRECCOVID | 72.89 | 77.03 | 78.95 | 73.25 | 79.58 | **86.11** |
| **Average** | 48.98 | 49.46 | 50.18 | 48.83 | 51.11 | **53.16** |

## B.3   Choice of In-Context Examples

Table 12 provides detailed numbers for varying in-context examples on all OOD BeIR tasks. Table 14 provides detailed numbers for various prompt formats on all OOD BeIR tasks.

Table 13: **In-Context Format** Comparing variants of in-context example format on *E5-Mistral-Instruct* during inference only. Training is done with the Regular format. Instruct refers to the baseline which does not use any in-context examples.

| Dataset | Instruct | RARe | | | | |
| | - | Query-Only | Doc-only | Shuffle-NC | Shuffle-C | Regular |
|---|---|---|---|---|---|---|
| ArguAna | **61.19** | 57.36 | 60.35 | 55.64 | 60.49 | 60.87 |
| ClimateFEVER | 39.03 | 38.35 | 38.32 | 37.44 | **37.84** | 37.50 |
| CQADupStack | 44.82 | 39.56 | 48.43 | 47.70 | 48.27 | **48.46** |
| DBPedia | 48.92 | 49.14 | 49.69 | 49.72 | **50.04** | 49.65 |
| FiQA2018 | 57.39 | 55.67 | 56.85 | 56.64 | **57.41** | 57.31 |
| NFCorpus | 40.99 | 41.00 | 42.09 | 42.02 | 41.92 | **42.28** |
| SCIDOCS | 17.94 | 19.06 | 20.06 | 19.98 | **20.25** | 20.19 |
| SciFact | 77.28 | 77.46 | 81.88 | 81.51 | 82.20 | **84.79** |
| Touche2020 | **29.35** | 27.04 | 29.02 | 28.60 | 29.31 | 28.70 |
| TRECCOVID | 72.89 | 75.11 | 79.97 | 79.07 | **80.03** | 79.58 |
| **Average** | 48.98 | 47.98 | 50.67 | 49.83 | 50.78 | **50.93** |

Table 14: **In-Context Format** Comparing variants of in-context example format on *E5-Mistral-Instruct*. Instruct refers to the baseline which does not use any in-context examples.

| Dataset | Instruct | RARe | | | | |
| | - | Query-Only | Doc-Only | Shuffle-NC | Shuffle-C | Regular |
|---|---|---|---|---|---|---|
| ArguAna | **61.19** | 58.88 | 57.54 | 60.17 | 58.97 | 60.87 |
| ClimateFEVER | **39.03** | 36.21 | 35.59 | 30.83 | 35.71 | 37.50 |
| CQADupStack | 44.82 | 46.66 | 48.28 | 45.78 | 47.97 | **48.46** |
| DBPedia | 48.92 | 49.98 | 49.08 | **50.93** | 50.24 | 49.65 |
| FiQA2018 | **57.39** | 54.44 | 56.02 | 54.25 | 55.98 | 57.31 |
| NFCorpus | 40.99 | 41.42 | 41.62 | 41.17 | 41.78 | **42.28** |
| SCIDOCS | 17.94 | 20.04 | 20.12 | 20.35 | 20.11 | **20.19** |
| SciFact | 77.28 | 78.84 | 79.80 | 80.70 | 80.51 | **84.79** |
| Touche2020 | **29.35** | 28.09 | 29.01 | 29.18 | 28.97 | 28.70 |
| TRECCOVID | 72.89 | 79.54 | **83.29** | 82.14 | 82.97 | 79.58 |
| **Average** | 48.98 | 49.41 | 50.04 | 49.55 | 50.32 | **50.93** |

### B.4 Mixture of Training Data

In Table 15, we analyze the impact of training with only in-context examples when starting from decoder-only LLMs. As opposed to starting from existing retriever models, which have been trained without in-context examples, we observe that performance drops in the instruction-only setting. This can be largely mitigated by considering a mixture of in-context and instruction-only queries.

### B.5 Applicability to Other Embedding Tasks

In Table 16, we report the performance on a subset of classification tasks from MTEB Muennighoff et al. (2023). We observe similar trends as retrieval, where providing in-context examples further enhances performance over providing task-specific instructions. The results underscore the applicability of RARe on embedding tasks beyond retrieval.

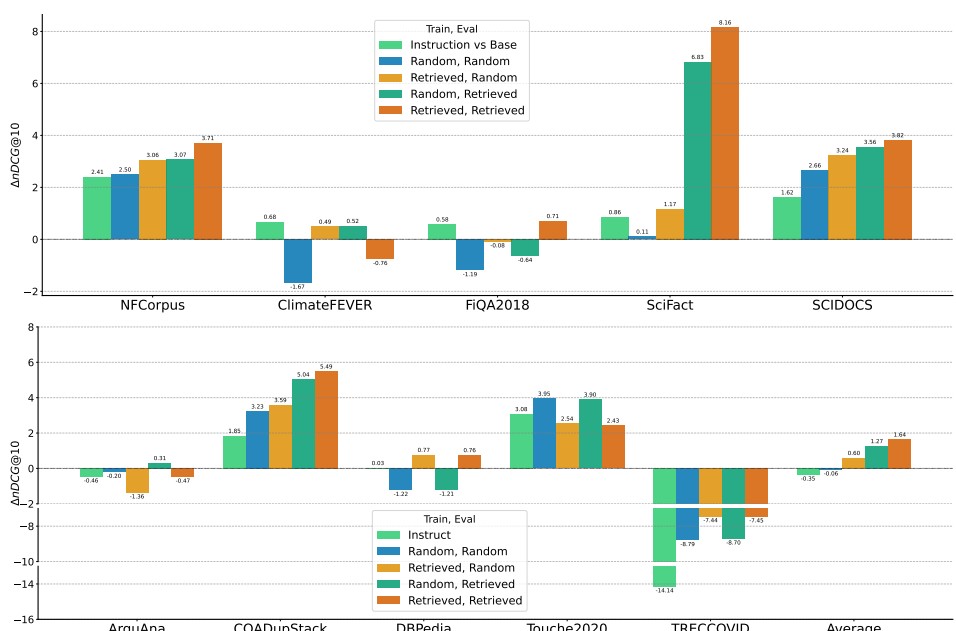

Figure 5: **Retrieved vs. Random In-context Examples.** Change in performance (ΔnDCG@10) on *E5-Mistral-Instruct* with RARe ($q^{\text{inst+ic}}$) from the baseline setting ($q$ both during training and evaluation time). Using retrieved examples during training and/or inference enhance model performance in 7/10 datasets.

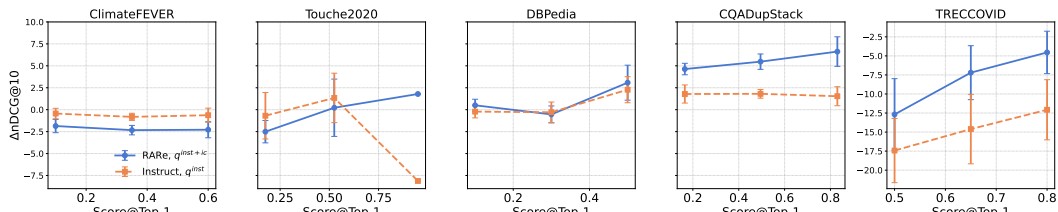

Figure 6: Change in performance (ΔnDCG@10) from the base model (*E5-Mistral-Instruct*) for varying similarity between the closest in-context example query and target query (Score@Top-1). Incorporation of semantically similar examples is beneficial on 3/5 datasets.

Table 15: Performance (nDCG@10) on datasets from the BeIR benchmark Thakur et al., 2021 when training decoder-only model (*Llama3*). Applying RARe with only in-context examples can lead to degradation of performance in the zero-shot setting ($q^{\text{inst}}$), but this is easily mitigated my including a mixture of $q^{\text{inst}}$ and $q^{\text{RARe}}$ data (30% and 70%) respectively.

| Training | Eval | NQ | Quora | NFCorpus | SciFact | SCIDOCS | FiQA2018 | CQA | Average |
|---|---|---|---|---|---|---|---|---|---|
| RepLLaMA-$q^{\text{inst}}$ | $q^{\text{inst}}$ | 62.70 | 88.34 | 37.73 | 75.02 | 19.66 | 44.31 | 42.59 | 52.91 |
| RARe-$q^{\text{RARe}}$ | $q^{\text{inst}}$ | 39.64 | **88.39** | 35.42 | 74.52 | **21.04** | 30.44 | 37.74 | 46.74 |
| | $q^{\text{RARe}}$ | 65.19 | 86.79 | 38.87 | **78.41** | 19.70 | 46.58 | **43.75** | 54.18 |
| RARe-$q^{\text{inst}} + q^{\text{RARe}}$ | $q^{\text{inst}}$ | 63.68 | 87.84 | 38.06 | 76.07 | 20.11 | 46.02 | 42.99 | 53.54 |
| | $q^{\text{RARe}}$ | **65.96** | 87.65 | **39.34** | 77.20 | 19.45 | **47.13** | 42.96 | **54.24** |

Table 16: Performance (accuracy) with *E5-Mistral-Instruct* on classification tasks, demonstrating the applicability of RARe on other embedding domains and tasks.

| Method | Banking77 | Emotion | Intent | ToxicChat | Avg |
|---|---|---|---|---|---|
| Base | 81.41 | 58.41 | 77.07 | 81.67 | 74.64 |
| Instruct | 83.27 | 58.00 | 77.57 | 82.31 | 75.29 |
| RARe | **86.98** | 57.18 | **78.11** | **83.99** | **76.57** |

| Method | DBPedia | NFCorpus | SciFact | Covid | Touche2020 |
|---|---|---|---|---|---|
| E5 (Wang et al., 2024a) | 40.7 | 35.0 | 70.4 | 74.1 | 30.9 |
| E5-Query2Doc (Wang et al., 2023) | 42.4 | 35.2 | 67.5 | 75.1 | 31.7 |
| E5-Mistral-Instruct (Wang et al., 2024b) | 48.92 | 40.99 | 77.28 | 72.89 | 29.35 |
| RARe | 49.65 | 42.28 | 84.79 | 79.58 | 28.7 |

Table 17: Comparisons against PRF method which use generative LLM to augment queries.

Table 18: **Comparison against PRF methods.** The RARe pipeline i.e. BM25 based in-context selector and LLM-based retriever significantly outperforms methods where a generative LLM is used as a rewriter and BM25 as a retriever (Lei et al., 2024; Jedidi et al., 2024).

| Method | Retriever | PRF /ICL Model | #Used Params | Arguana | Covid | FiQA | SciFact | DBPedia | NFCorpus |
|--------|-----------|----------------|--------------|---------|-------|------|---------|---------|----------|
| ReDE-RF | BM25 | Mistral-7B | 7B | - | 65.60 | 29.30 | 66.90 | 37.60 | 35.50 |
| CSQE | BM25 | ChatGPT-3.5 | Unknown | 40.30 | 74.20 | 25.00 | 69.60 | 40.30 | - |
| RARe | E5-Mistral-7B | BM25 | 7B | **60.87** | **79.58** | **57.31** | **84.79** | **49.65** | **42.28** |

Table 19: Performance of RARe against Instruct variant trained with 5× steps. RARe performs better, especially on OOD settings, while Instruct begins to overfit on the in-doomain data.

| Method | BeIR | | RAR-b |
|---|---|---|---|
| | ID | OOD | |
| Instruct | 72.91 | 48.98 | 24.12 |
| Instruct (5x Steps) | 72.97 | 48.04 | 24.02 |
| RARe (5 examples) | 72.98 | 50.93 | 25.79 |

