# OpenReview forum: "RARe: Retrieval Augmented Retrieval with In-Context Examples"
_colmweb.org/COLM/2025/Conference — COLM 2025_

### Official Review · Reviewer_9t1y · 2025-05-11

**Rating:** 5
**Confidence:** 5
**Ethics Flag:** 1

**Summary:**

This paper proposes a ranker, which I believe is meant to be a first-stage retriever, although the paper neglects to state whether it is a first-stage retriever or a reranker. This ranker augments the query with an instruction, and related queries and relevant documents. For training and most of the experiments, it used 5 examples in the expanded query.

The paper shows that in order to cope with this expanded query, the model has to be trained for it. Fine-tuning Llama-3 is better than fine tuning Llama-3.1-Instruct for MS MARCO. However Llama-3.1-Instuct is slightly better for BEIR and much better for RAR-b. When comparing to alternative models, Prompriever is competitive except for RAR-b. Given that RAR-b is not retrieving documents, Bright might be a better comparison for out-of-domain.

The paper also reports results on fine-tuning two retrieval models - LLM2Vec-Llama-3-8B-Supervised and E5-Mistral-Instruct. Here results are only reported on BEIR and RAR-b. It is unclear why MS MARCO is excluded. The results are mixed relative to starting with models that are not Retriever baselines on the Llama family; however, the best performance is achieved on BEIR and RAR-b using E5.

Several ablations are performed. The first examines the importance of the example queries being related to the test query or the documents being related to the query. The setting where the query and documents are related to the test query is reminiscent of search engines fixing the results of popular queries based on what others liked to see for that query. While random queries and random documents were the least helpful, having only similar documents was nearly as helpful as having both similar queries and similar documents. This indicates that the technique is gaining benefits similar to the say that pseudo relevance feedback. While there is a comparison with PRF, it is a BM25-based PRF. Given that there a dense retrieval versions of PRF, comparing to a BM25 approach is not very informative.

The paper could be improved by situating it better in the context of other work on retrieval. It would also benefit from having more consistency in terms of evaluation datasets and the information included with respect to results. For instance Table 1 has a training data column, but Table 2 does not, why? Table 2 has an "instruct" row, but Table 1 does not, why?

While it is generally easy to understand what the paper has done, it is more challenging to place the work in the context of the work that has come before. Exploring new ways to construct prompts for retrieval has merit; however, it must be acknowledged that first stage retrieval must be very fast.  It may be that greater computational effort can only be afforded on a relatively small subset of documents. Such an expensive approach might be better evaluated as a re-ranker.

**Questions To Authors:**

Q1: how does this approach perform on Bright?

Line 42: new query format -> new query formats
It seems like the Inference-only Modification should be Section 5.1 since it is strange for like 143 to refer to Section 5, meaning the Inference-only Section.

**Reasons To Accept:**

+ Reimagining pseudo relevance feedback in a prompt based approach is an interesting direction
+Innovative approach to prompt-based retrieval

**Reasons To Reject:**

- It is questionable whether the differences in performance are actually meaningful, especially between Promptrieve and RAR-e when the same base model is fine-tuned. Statistical tests would be helpful
- The paper lacks clarity when positioning the work in the context of other work that has come before, particularly on the type of retrieval and query augmentation
- The latency introduced by such a long prompt is only acknowledged in the appendix where in some cases the latency increase is 40 times larger than the original query. Given the expectation that search is nearly instantaneous, even doubling the latency is likely unacceptable.

---

> ### Author Response · Authors · 2025-06-02
>
> Thank you for your review! We address your comments below.
>
> > **Unclear why MS-MARCO has been excluded**
>
> We do not exclude MS-MARCO for any experiment. For LLM2Vec-Llama-3-8B-Supervised and E5-Mistral-Instruct, MS-MARCO is part of the In-Domain split of BEIR (among 4 other datasets). See Table 6 in the Appendix for individual numbers on MS-MARCO on these models.
>
> When we train on MS-MARCO, all other datasets in BEIR are out-of-domain, which is why we report MS-MARCO separately for those experiments.
>
> We will update Tables 1 and 2 to enhance clarity.
>
> > **Given that there a dense retrieval versions of PRF, comparing to a BM25 approach is not very informative.**
>
> Below, we provide a comparison against Query2Doc [1]. This is a PRF method which uses a  proprietary generative LLM to augment queries and a fine-tuned dense retriever instead of BM25 for embeddings.
>
> Method             | DBPedia | NFCorpus | SciFact | Covid | Touche2020 |
> | --- | --- | --- | --- | --- | --- |
> E5                     | 40.7 | 35.0 | 70.4 | 74.1 | 30.9 |
> E5-Query2Doc | 42.4 | 35.2 | 67.5 | 75.1 | 31.7 |
> E5-Mistral-Instruct       | 48.92 | 40.99 | 77.28 | 72.89 | 29.35 |
> RARe               | 49.65 | 42.28 | 84.79 | 79.58 | 28.7 |
>
> We see that RARe offers higher improvements on average compared to Query2Doc over their respective baselines.
>
> [1] https://aclanthology.org/2023.emnlp-main.585.pdf
>
> > **Table 1 has a training data column, but Table 2 does not, why?**
>
> We apologize for the confusion. We aimed to highlight the main aspect that we were varying/studying in each table. All models in Table 2 use the same training dataset. In Table 1, Promptriever uses an additional synthetically augmented dataset (Line 170) from GPT-4o. We will make this clearer in the revision.
>
> > **Table 2 has an "instruct" row, but Table 1 does not, why?**
>
> For Table 1, the baseline RepLLaMA is the equivalent of “Instruct”. Since RepLLaMA only trains on a single task MSMARCO, it provides null string as the as instruction (Line 171).
>
> > **Latency**
>
> As shown in Table 10, the cost of constructing in-context examples is minimal, since we retrieve from the queries in the training/validation set rather than the full retrieval corpus.
> The primary source of latency arises from encoding longer queries by the LLM - a limitation that is inherent to any PRF-based method involving query expansion. Compared to other query augmentation approaches such as ReDe-RF [1], CSQE [2], or HyDE [3], which involve generating hypothetical documents using LLMs, RARe may also be more efficient as it only requires a single forward pass through the LLM.
>
> In our current implementation, no additional optimizations such as prompt caching are applied. With such optimizations in place, we expect the additional latency of query encoding to also be minimized. Moreover, the cost of encoding longer queries is typically outweighed by the time required for similarity search using FAISS over large indexes. For example, the 40x latency on NFCorpus is an artifact of a very small index (ony ~3000 documents to retrieve from). We acknowledge that efficiency wise, ‘Instruct’ may be a better choice over RARe (or any other PRF or query expansion method) when the document index is very small.
>
> [1] https://arxiv.org/abs/2410.21242
>
> [2] https://arxiv.org/abs/2402.18031
>
> [3] https://aclanthology.org/2023.acl-long.99/
>
> > **Statistical tests would be helpful**
>
> We provide statistical significance tests for each individual dataset in the Appendix (Tables 6–9) via Fisher’s method. * indicates p<0.05.
>
> RARe is not consistently superior to Promptriever on BEIR, which benefits from additional synthetic data generated using GPT-4o, but achieves statistically significant improvements on 8 out of 15 BEIR datasets: CQADupStack, FiQA, NFCorpus, HotpotQA, MS MARCO, Natural Questions, TREC-COVID, and Touche2020. When compared to RepLLaMA, RARe demonstrates statistically significant gains on 11 out of 15 datasets.
> On RAR-b, RARe significantly outperforms Promptriever on aNLI, PiQA, TempReason, and WinoGrande (Table 9).

---

> > ### Comment · Reviewer_9t1y · 2025-06-07
> >
> > Thank you for responding to my review.
> >
> > In reviewing my confusion about MSMARCO, I believe it came from inconsistent headers in Table's 1 and 2 with ID/OOD as the high level in T1 but the lower level in T2. Making these two tables as similar as possible would assist the reader in understanding what you are reporting.

---

> > > ### Author Response · Authors · 2025-06-08
> > >
> > > Thank you for your feedback. We agree that the headers in Tables 1 and 2 should be more consistent, and we will revise them in the updated draft to follow a common structure, as you suggested.
> > >
> > > If you have any remaining concerns, we would be happy to address them. Otherwise, if you feel your concerns have been fully resolved, we would be grateful if you would consider updating your score.

---

> > > > ### Comment · Reviewer_9t1y · 2025-06-09
> > > >
> > > > Thank you for addressing that concern.

---

### Official Review · Reviewer_1TMR · 2025-05-12

**Rating:** 6
**Confidence:** 5
**Ethics Flag:** 1

**Summary:**

The authors finetune embeddings for retrieval with demonstrations, then use demonstrations during inference to improve retrieval results. It is very similar to BGE-EN-ICL (https://arxiv.org/abs/2409.15700) although with more analysis and different approach to selecting demonstrations.

**Questions To Authors:**

It seems strange to use BM25 for demonstration retrieval. Why not use a dense embedding?

Did you consider using a HyDE-like approach? This seems very well suited for few-shot.

**Reasons To Accept:**

A1. The authors explore the idea of ICL for embeddings for document retrieval, which is a more unusual setting for ICL. It remains unclear to me if what we are seeing is ICL or simply query expansion, and I suspect it is the latter.

A2. The authors explored various model settings, often different versions of same model (llama 2 vs 3 vs LLM2Vec vs Promptriever; also SFR mistral vs E5-mistral). Also they explore the highly studied BEIR dataset (small improvement), and a relatively new and challenging dataset for retrieval + reasoning (RAR-b) (moderate improvement).

A3. There are various ablations validating design choices including random vs retrieved demonstrations, number of demonstrations, format, inclusion of negative documents.

**Reasons To Reject:**

R1. The main reason to reject is this work is so similar to BGE-EN-ICL (Li et al., Sep 2024. "Making Text Embedders Few-Shot Learners") which is already a popular embedding and paper. This work is hardly discussed and at minimum it seems it should be a baseline. If the baseline is not valid from data perspective, then attempt to reproduce their training seems reasonable instead.

R2. The other fairly large reason to reject is that it seems very likely this is just query expansion and not ICL at all. a. We would expect that query expansion is effective, and b. We would also expect that dense embeddings without additional training may struggle with long contexts caused by including demonstrations. This deserves much more focus in this paper and simply saying "One can view in-context examples as a form of query expansion" is not sufficient. Examples of how to show this is not just query expansion at work is to try the same approach with techniques like BM25 and see if there is comparable improvement, but this is just one idea.

R3. It's hard to follow the results in the paper. For example, Figure 2 does not contain any RARe and is the only place where SFR is included, and Table 4 includes text-embedding-3-large but it's never mentioned in the text. It's also not clear in Table 1 if the RepLlama and Promptriever baselines were trained as part of this paper, and whether RARe is using demonstrations in Table 1 (I think not?).

R4. (low-ish priority) There is not much novelty in the training approach. The authors simply sample few-shot demonstrations during SFT w/ constrastive embedding training. It does seem like the right starting point, but I would expect that various iterations on this setup would help, e.g. using RL or having a joint objective focused on generation.

R5. (low-ish priority) It's not immediately clear to me what is the utility of embeddings that can do ICL. Embeddings can be finetuned fairly quickly on 10k+ samples, and if you have demonstrations already then synthetic data is fairly easy to generate (see Promptagator). I would guess simply SFT with domain-specific synthetic data will probably work better than ICL plus be more robust to various hyperparams.

---

> ### Author Response · Authors · 2025-06-02
>
> Thank you for your review! We address your comments below.
>
> > **Discussion w.r.t BGE-ICL**
>
> Thank you for the comment! We discuss our work in relation to this work below. We agree with the reviewer about a comparison of the method. Our **Random-Random** baseline in Figure 3 is the same as the training pipeline of BGE-ICL with two minor differences: (1) The training data, and (2) their approach uses a variable number of ICL examples during training.
>
> We highlight two new contributions:
>
> 1. While both work explores in-context examples for encoding queries, our work further proposes *retrieving* semantically similar in-context examples. We empirically show semantically similar examples are beneficial, whether used during training, inference, or both (Figure 3).
> 2. Detailed analysis of the method:
>
>      (i) We demonstrate that this can be applied to both LLM and retriever checkpoints.
>
>      (ii) We present an ablation to study whether ICL format matters or simply query expansion helps. This is shown in Table 4, where we compare against various baselines (shuffled, queries-only, etc).
>
>      (iii) Analogous to the role of negative examples in contrastive learning, we investigate whether incorporating negative docs into the context yields performance gains. Our findings indicate that such inclusion offers no substantial advantage over only positive docs (Table 5).
>
>      (iv) We include an efficiency analysis, breaking down the contribution of each component (Table 10).
>
>
> Our work was developed concurrently but as Li et al is already accepted to a conference, we will revise our draft to re-position our paper accordingly. While it is hard to argue using in-context examples for retrieval as our novel contribution, our work significantly deepens the study of using in-context examples in retrieval models.
>
>
> > **Query-Expansion vs ICL & BM25**
>
> Thanks for thoughtful feedback! We agree understanding whether adding in-context examples is merely query expansion or not is important to understand.
>
> We compare RARe with ReDe-RF [1] and CSQE [2]: both PRF (Pseudo Relevance Feedback)-based query augmentation approaches (extensions of HyDE), in Table 17 of the Appendix.
>
> We additionally tried expanding the queries to BM25 with the same demonstrations - i.e. we augment retrieved in-context examples at test time to the query. We provide results in the table below. Note that since BM25 is a sparse retriever, it cannot be trained with our approach, rather only an inference-time modification can be made, similar to the experiments in Figure 2.
>
> We observe that performance drops when using BM25 with retrieved in-context examples, similar to the degradation seen with inference-only modifications in dense retrievers in our experiments (Figure 2). This highlights several key points:
> 1. The assumption that performance degradation in dense retrievers under inference-only modification is due to longer input context may not be true, since even sparse retrievers like BM25 exhibit a similar drop.
> 2. If the benefits were simply due to query expansion, BM25 should have improved, as it does with ReDe-RF [1]/CSQE [2], which works out of the box for sparse retrievers.
> 3. This underscores the importance of our approach, where training with in-context examples is critical. It enables the model to learn how to utilize relevant signals and mappings from the Query-Document pairs in context effectively.
> 4. Our experiments in Table 4 additionally disentangle the effects of expanding the query from following the ICL format, by training and evaluating with alternative formats. Here, the best performance is achieved with the full ICL format, further indicating that the improvements cannot be explained by query expansion alone (shuffled setting, despite having the same content but trained and evaluated with the shuffled format does not perform as well).
>
> | Dataset | BM25 | + IC (Inference-Only Modification) |
> |---|:----:|:----:|
> | ArguAna | 49.27 |  32.39 |
> | ClimateFEVER | 13.62 |  9.36 |
> | CQADupStack  | 31.86 | 18.25 |
> | DBPedia | 29.91 | 11.83 |
> | FEVER | 48.10 | 5.48 |
> | FiQA2018 | 25.14 | 2.16 |
> | HotpotQA | 56.91 | 24.73 |
> | MSMARCO | 21.88 | 3.75 |
> | NFCorpus | 32.08 | 18.79 |
> | NQ | 28.51 | 25.67 |
> | Quora | 80.42 | 27.53 |
> | SCIDOCS | 15.78 | 11.32 |
> | SciFact | 68.79 | 42.02 |
> | Touche2020 | 33.05 | 39.43 |
> | TRECCOVID | 62.31 | 39.42 |
>
>
> [1] https://arxiv.org/abs/2410.21242
>
> [2] https://arxiv.org/abs/2402.18031

---

> > ### Comment · Reviewer_1TMR · 2025-06-02
> >
> > Quick follow up to address one thing:
> >
> > > concurrent work
> >
> > ICML has this guideline re: concurrent work (https://icml.cc/Conferences/2025/ReviewerInstructions):
> >
> > > Authors cannot expect to discuss other papers that have only been made publicly available within four months of the submission deadline. (This cut-off is adopted from the AISTATS and ICLR reviewing instructions.) Such recent papers should be considered as concurrent and simultaneous. Good judgement is necessary to decide whether a paper that has not yet been peer-reviewed should be discussed. The guideline is to follow the best practices of the specific subfield; the Area Chair can help in each case with these.
> >
> > Given that BGE-ICL appeared more than four months before COLM submission deadline (March 2025), then it seems like BGE-ICL should be discussed just like any other related work. Can you explain more clearly how you will revise your draft to re-position your paper.

---

> > > ### Author Response · Authors · 2025-06-03
> > >
> > > Thank you for your feedback.
> > >
> > > To better position our paper and clarify its contributions, we will revise the draft in the following ways:
> > >
> > > 1. We will include the methodological equivalent of BGE-ICL in Tables 1 and 2 to provide a direct, quantitative comparison. We will additionally discuss BGE-ICL in Section 3 (Method).
> > >
> > > 2. We will revise our contribution statements to more accurately reflect the key findings and insights of the paper. In particular, we will emphasize:
> > >     * Our proposal to retrieve semantically similar in-context examples for improving in-context learning (ICL) in embedding-based retrievers.
> > >
> > >     * A detailed empirical study examining key design choices for in-context examples, such as their quality, quantity, choice of base model, and format, in order to better understand the source of performance gains in embedding-based ICL.
> > > 3. We will rewrite the related work section to position our study with respect to BGE-ICL, clarifying the similarities and differences outlined above.
> > >
> > > We are happy to address any further questions you may have.

---

> > > > ### Comment · Reviewer_1TMR · 2025-06-03
> > > >
> > > > Thank you for the quick response. I would say this satisfies any concern I had about overlap with BGE-ICL.
> > > >
> > > > I am leaning more positive.

---

> > > > > ### Author Response · Authors · 2025-06-04
> > > > >
> > > > > We're glad our clarifications addressed your concerns. If you feel all of your concerns have been resolved, we’d be grateful if you could consider adjusting your score to reflect your updated evaluation.
> > > > >
> > > > > We are happy to address any other questions or comments you may have.

---

> ### Author Response · Authors · 2025-06-02
>
> We address additional comments below
>
> > **RepLlama and Promptriever baselines**
>
> We acknowledge that Table 1 could be clearer regarding the training setup for the baselines and RARe’s use of demonstrations.
>
> RepLLaMA was re-trained using the publicly available implementation provided by its authors, as the paper originally used Llama-2. The best performing Promptriever models (which use more recent Llama-3) were already hosted on huggingface, which we evaluated on with dataset-specific prompts provided in the official code.
>
> As for RARe, it does use demonstrations in Table 1 (5 examples, as noted in Line 135).
>
>
> > **ICL vs. Fine-tuning**
>
> We agree that when thousands of demonstrations are available, fine-tuning may be more effective.
>
> The advantage of RARe is the ability to adapt in a few-shot manner to new tasks or domains at test time via a small number of examples. RARe, even when trained with the Retrieved format and evaluated with Random (Retrieved-Random), still benefits over Random-Random and Instruct (Figure 3).
>
> This shows that unlike fine-tuning approaches, RARe does not require generating larger amounts of synthetic data or training the model on new tasks, providing the user with fine-grained control of the retrieved documents by providing representative demonstrations at inference time. Moreover, RARe does not rely on domain-specific hyperparameter tuning other than number of few-shot examples.
>
> > **BM25 vs Dense**
>
> We empirically observed that BM25 is lightweight yet effective. As shown in Table 10 Appendix, the primary contributor to latency is the query encoding step performed by the dense encoder. Since BM25 performed reasonably well in our experiments, we opted to use it throughout.
>
> > **RL and Joint Objective For Generation &  HyDE-like Approach**
>
> This is indeed an interesting direction to pursue! Retrievers that first generate and then embed potentially trained using a reinforcement learning or joint generative objective could combine the strengths of both generative models (used in HyDE-like approaches) and dense retrievers. While exploring this approach is beyond the scope of the current work, it presents a promising avenue for future research.

---

### Official Review · Reviewer_neiN · 2025-05-12

**Rating:** 5
**Confidence:** 5
**Ethics Flag:** 1

**Summary:**

This document explores RARe: Retrieval-Augmented Retrieval With In-Context Examples,  which fine-tunes pre-trained models using in-context examples whose queries are semantically similar to the target query. This method, applied to both decoder-only language models and existing retriever models, demonstrates significant performance gains, particularly in out-of-domain generalization. The paper also analyzes factors influencing performance, such as the selection, quantity, and format of in-context examples.

**Reasons To Accept:**

* In context learning has been shown effective for improving large language model predictions.  RARe brining ICL to dense retrieval models is very intuitive.
* Extensive ablation studies to show the effectiveness of the proposed approach, as well as analysis on the impact of model quality by various factors, e.g., number of examples, random vs retrieved examples, prompt format.
* The paper is generally well written and easy to follow.

**Reasons To Reject:**

* The main concern is the similarity with  Li et al., 2024.   Li et al., 2024 presented almost the same approach and was first seen Sep-2024 and has seen be cited 20+ times.   The authors referred Li et al., 2024 as concurrent work, which isn't fully comply with ARR citation policies [2] which states `papers (whether refereed or not) appearing less than 3 months before the submission deadline should be considered contemporaneous to the submission`.
* For OOD dataset, examples are synthetically generated. The paper mentions that gains with increasing Score@top-1 are less pronounced one OOD Datasets.  How does the quality of the synthetically generated questions affect RARe performance?   Should this analysis potentially provides practitioners a guidance on when to use RARe vs Instruct?

* [1] https://arxiv.org/abs/2409.15700
* [2] https://www.aclweb.org/adminwiki/index.php/ACL_Policies_for_Review_and_Citation

---

> ### Author Response · Authors · 2025-06-02
>
> Thank you for your review! We address your comments below.
>
> > **Similarity with Li et al, ICLR 2025.**
>
> Thank you for the comment! We discuss our work in relation to this work in detail below. You are correct that our method (RARe) is very similar to Li et al (2024). Our **Random-Random** baseline in Figure 3 is the same as the training pipeline of BGE-ICL with two minor differences: (1) The training data, and (2) their approach uses a variable number of in-context examples during training.
>
> We highlight two new contributions of our work:
>
> 1. While both work explores in-context examples for encoding queries, our work further proposes *retrieving* semantically similar in-context examples. We empirically show semantically similar examples are beneficial, whether used during training, inference, or both (Figure 3).
> 2. Detailed analysis of the method:
>
>       (i) We demonstrate that this can be applied to both LLM and retriever checkpoints.
>
>       (ii) We present an ablation study to study whether ICL format matters or simply query expansion helps. This is shown in Table 4, where we compare against various baselines (e.g., shuffled setting, queries-only setting, etc).
>
>      (iii) Analogous to the role of negative examples in contrastive learning, we investigate whether incorporating negative documents into the in-context learning (ICL) context yields performance gains. Our findings indicate that such inclusion offers no substantial advantage over using only positive documents (Table 5).
>
>      (iv) We include an efficiency analysis, breaking down the contribution of each component (Table 10).
>
> Our work was developed concurrently but as Li et al is already accepted to a conference, we will revise our draft to re-position our paper accordingly. While it is hard to argue using in-context examples for retrieval as our novel contribution, our work significantly deepens the study of using in-context examples in retrieval models.
>
> > **For OOD dataset, examples are synthetically generated. How does the quality of the synthetically generated questions affect RARe performance?**
>
> We clarify that among the OOD datasets: NFCorpus, FiQA-2018, SciFact, and DBPedia do not use synthetically generated examples, rather their original training splits. For the rest (ArguAna, ClimateFEVER, Covid, CQADupStack, SCIDOCS, Touche2020), synthetic examples used are provided by BEIR and were not generated by us. We will include these details in the Appendix of the updated draft.
>
> While the quality of synthetically generated examples may influence performance, we did not observe any consistent correlation between performance and whether the examples were synthetic. Investigating this further is indeed an interesting direction but is beyond the scope of this work.

---

> > ### Comment · Reviewer_neiN · 2025-06-04
> >
> > Thanks for the response.
> >
> > On the one hand, retrieved in-context examples perform much better than random ones.
> > On the one hand, the quality of synthetically generated queries in the in-context examples do not have a significant impact on model performance.
> > Since using semantically similar examples is one of the main contribution compared to BEG-ICL, further explanation is needed.

---

> > > ### Author Response · Authors · 2025-06-04
> > >
> > > Dear Reviewer neiN,
> > >
> > > Thank you for your feedback. We would appreciate it if you could elaborate on the specific aspects where you believe further explanation is needed.
> > >
> > > We will update the draft with the following changes that distinguishes our work from BGE-ICL.
> > > 1. We will include the methodological equivalent of BGE-ICL in Tables 1 and 2 to provide a direct, quantitative comparison. We will additionally discuss BGE-ICL in Section 3 (Method).
> > > 2. We will revise our contribution statements to more accurately reflect the key findings and insights of the paper. In particular, we will emphasize:
> > >     (i) Our proposal to retrieve semantically similar in-context examples for improving in-context learning (ICL) in embedding-based retrievers.
> > >     (ii) A detailed empirical study examining key design choices for in-context examples, such as their quality, quantity, choice of base model, and format, in order to better understand the source of performance gains in embedding-based ICL.
> > >
> > > We are happy to further address any concerns you may have.

---

### Official Review · Reviewer_cnSt · 2025-05-14

**Rating:** 5
**Confidence:** 3
**Ethics Flag:** 1

**Summary:**

The paper looks at dense retrieval using two-tower model. They propose to augment the query with query-document pairs where the similar queries are retrieved by using BM25. They did a detail study on how the method is working with different setup, such as fine-tuned or not fine-tuned, different checkpoints, etc.

**Questions To Authors:**

There seems to be a few dense retrieval papers prior to 2020 that were not cited, such as the Universal Sentence Encoder (Cer et al., 2018), Sentence Embeddings using Siamese BERT-Networks (Reimers and Gurevych 2019).

**Reasons To Accept:**

Exploring in-context examples for the retrieval setting is interesting.

**Reasons To Reject:**

I have a few questions in terms of baselines. I'm wondering if how does the proposed methods compared to pre-pending the query with in-context queries instead of examples. Also, how does this method compare to BM25 with original query as a baseline? Does the performance gain justify the cost of constructing the in-context examples by retrieval?

---

> ### Author Response · Authors · 2025-06-02
>
> Thank you for your review! We address your comments below.
>
> > **I'm wondering if how does the proposed methods compared to pre-pending the query with in-context queries instead of examples.**
>
> Thanks for the suggestion! We perform careful ablation and present the result in Table 4. The experiment you are asking for is represented as the “Queries-Only” row in Table 4. This provides only in-context queries, rather than full examples. We find this underperform base Instruct or the full ICL format on average. We copy-and-paste results here again for convenience.
>
> Method | Avg nDCG |
> | --- | --- |
> Instruct |  51.83 |
> Queries-Only | 51.39 |
> RARe (Regular) | 53.74 |
>
>
> > **Also, how does this method compare to BM25 with original query as a baseline?**
>
> All of our LLM-based dense retriever baselines outperform BM25 as a retrieval method. We provide a comparison below.
>
> | Dataset      |  BM25 | E5-Mistral-Instruct |
> |--------------|:-----:|:-------------------:|
> | ArguAna      | 49.27 |        61.19        |
> | ClimateFEVER | 13.62 |        39.03        |
> | CQADupStack  | 31.86 |        44.82        |
> | DBPedia      | 29.91 |        48.92        |
> | FEVER        | 48.10 |         91.5        |
> | FiQA2018     | 25.14 |        57.39        |
> | HotpotQA     | 56.91 |        73.91        |
> | MSMARCO      | 21.88 |        41.89        |
> | NFCorpus     | 32.08 |        40.99        |
> | NQ           | 28.51 |        67.44        |
> | Quora        | 80.42 |        89.82        |
> | SCIDOCS      | 15.78 |        17.94        |
> | SciFact      | 68.79 |        77.28        |
> | Touche2020   | 33.05 |        29.35        |
> | TRECCOVID    | 62.31 |        72.89        |
>
>
> Please also refer [1] and [2], which show that our baseline E5-Mistral > Contriever > BM25 on the BEIR benchmark.
>
> [1] https://arxiv.org/pdf/2401.00368
>
> [2] https://arxiv.org/pdf/2112.09118

---

> > ### Comment · Reviewer_cnSt · 2025-06-10
> >
> > Thank you for addressing the questions and for pointing out the results.
> >
> > I agree the prepending in-context examples retrieved by semantically similar queries in the training corpus is an interesting method for dense retrieval. The concern would be the additional encoding cost at inference time.

---

### Decision · Program_Chairs · 2025-07-07

**Decision:**

Accept

**Comment:**

This paper looks at In-Context Examples for Decoder only LLMs for IR. Overall, the reviewers were rather close to borderline without any being significantly excited. There were multiple questions about table headers (1 & 2) that were addressed, as well as other clarification questions. There was lots of discussion in the rebuttal period with authors and reviewers that assuaged many of these issues. However, most of the reviewers kept a score of 5 (marginally below acceptance). In part, this had to do with similarity to the BEG-ICL paper which appeared 4 months before the COLM deadline (with 3 months being concurrent work definition). The authors addressed the difference of their approach in the discussion period (semantically-related ICL examples vs. random fixed set in BEG-ICL). In addition, most of the reviewers had other small concerns (i.e., latency, encoding cost at inference time). Coupled with the smaller concerns, and the fact that the paper did not address BEG-ICL initially (which meant much of the discussion period was focused on this), there were no strong champions among the reviewers to accept the paper.

I suspect reviewers would have been more positive had the paper discussed BEG-ICL initially, so it seems mostly a related work issue, which the authors say they will address in the next version of the paper. The authors put a lot of effort into rebuttals and likely would have swayed reviewers on many other points if not for these initial concerns.

It's critical the authors discuss recent related work (BEG-ICL) in their revision of the paper.